# Changes in Phytochemical Content, Antioxidant Activity, and Anti-Inflammatory Properties of *Cudrania tricuspidata* Fruits Treated by Roasting

**DOI:** 10.3390/foods12112146

**Published:** 2023-05-26

**Authors:** Si Young Ha, Ji Young Jung, Jae-Kyung Yang

**Affiliations:** Department of Environmental Forest Science, Institute of Agriculture and Life Science, Gyeongsang National University, Jinju 52828, Republic of Korea; hellohsy2@gmail.com (S.Y.H.); jungjy@gnu.ac.kr (J.Y.J.)

**Keywords:** antioxidant activity, anti-inflammatory, roasting, *Cudrania tricuspidata*, flavan-3-ols, phytochemical

## Abstract

The study investigated the antioxidant effects of roasted *Cudrania tricuspidata* (*C. tricuspidata*) fruits by comparing them with unroasted *C. tricuspidata* fruits. The results showed that the roasted *C. tricuspidata* fruits (150 °C, 120 min) exhibited significantly higher antioxidant activity, especially in terms of anti-inflammatory effects, than the unroasted fruits. Interestingly, there is a high correlation between the color of the roasted fruit and the antioxidant activity. Heating disrupts cells and deactivates endogenous oxidative enzymes, leading to an increase in flavonoid content. Moreover, heat treatment may also interfere with plant metabolism, thereby influencing flavonoid content. Moreover, an HPLC analysis of roasted fruits in our study showed that the increase in antioxidant activity was attributed to the increase in flavan-3-ols and phenolic acids in the roasted *C. tricuspidata* fruits. To the best of our knowledge, this is the first time the antioxidant activity and anti-inflammation of roasted *C. tricuspidata* fruits was studied. The study concluded that roasted *C. tricuspidata* fruits could be a valuable natural source of antioxidants for various food and medicinal applications.

## 1. Introduction

Synthetic antioxidants commonly used in various applications, such as butylated hydroxyanisole and butylated hydroxytoluene, exhibit exceptional antioxidant properties. However, due to their potential carcinogenicity and allergenicity, these substances are subject to stringent regulations [1]. As a result, researchers have increasingly directed their efforts towards identifying safer and natural alternatives to these chemicals [2]. In recent years, considerable attention has been given to exploring the potential of herbal medicines and edible plant extracts as sources of natural antioxidants [3]. Phenolic compounds, predominantly present in different plant parts such as flowers, leaves, and bark, are believed to offer remarkable antioxidant effects with enhanced safety profiles [4]. Therefore, the objective of this study was to investigate the antioxidant activity of roasted *Cudrania tricuspidata* fruits, aiming to identify them as a potential source of natural antioxidants for diverse food and medicinal applications.

*Cudrania tricuspidata* (*C. tricuspidata*) is a deciduous tree primarily found in the hilly regions of East Asia, including Korea, Japan, China, and eastern Russia [5]. Traditionally, this tree has been used for medicinal purposes, treating conditions such as eczema, pulmonary tuberculosis, and acute arthritis [6]. Recent investigations have highlighted the antibacterial properties and ability to inhibit peroxidative lipid production in the leaves, stems, and roots of the mulberry tree [7]. The fruit of the mulberry tree, known for its high content of polyphenols—a crucial indicator of antioxidant activity—is extensively utilized in food ingredients and private-sector cancer treatments [8]. Methanol and ethanol extracts of mulberry fruits exhibit substantial levels of polyphenols and flavonoids, which demonstrate potent antioxidant effects [9]. Furthermore, the crude polysaccharides found in *C. tricuspidata* fruit have been found to activate antioxidants and possess neuroprotective effects [10]. Nevertheless, due to its limited storage life, additional processing methods are necessary to prolong the fruit’s shelf life. Heat treatment is a widely employed method in food processing to improve food quality and extend its shelf life [11]. Nevertheless, this process has certain drawbacks, such as the degradation of heat-sensitive nutrients and loss of active compounds. Recent investigations have been carried out to address these limitations, examining the changes that occur in fruits and vegetables when subjected to various heat treatments, including blanching at approximately 100 °C, dry heat treatment such as roasting at 150–200 °C, and wet heat treatment such as high-pressure sterilization at 120 °C [12]. Additionally, extensive research has been conducted to enhance the antioxidant activity of diverse fruits and vegetables, including ginseng, garlic, apples, and watermelons [13]. Studies have indicated that heat treatment enhances the antioxidant activity of tomatoes, shiitake mushrooms, sweet corn, and citrus peels [14]. Despite the widespread use of *C. tricuspidata* as both a medicinal plant and food ingredient, research focusing on the antioxidant and anti-inflammatory effects of heat treatment remains limited. Hence, the present study aims to optimize the heat treatment conditions to maximize the antioxidant activity of *C. tricuspidata* and explore its potential anti-inflammatory effects.

## 2. Materials and Methods

### 2.1. Fruit Samples and Heat Treatment

*Cudrania tricuspidata* fruits were obtained from a pesticide-free farm in Andong-si, Gyeongsangbuk-do, South Korea. The *C. tricuspidata* were ascertained by the experts of Korea National Arboretum and a voucher specimen (Accession no: CJUA200909151089) was deposited for this collection. Damaged fruits were excluded, and heat-treated fruits were dried at room temperature for 24 h prior to use. Heat treatment was performed using a roasting device (CBR-101, Gene Cafe, Ansan, Republic of Korea) to evaluate the effects of different roasting conditions on antioxidant activity. Roasting was conducted at temperatures ranging from 110–150 °C for 15–120 min. The roasted fruits were then frozen at −40 °C for 24 h, followed by freeze-drying for 72 h (FD5508 Bondiro, Ilshin, Siheung, Republic of Korea). The dried fruits were ground into a powder using a grinder and sieved through a 20 mesh pass 80 mesh standard body.

### 2.2. Measurement of Color

Color attributes of the roasted fruits, including L* (lightness/darkness), a* (redness/greenness), and b* (yellowness/blueness), were evaluated using a colorimeter (CR-20; KONICA MINOLTA, Tokyo, Japan). The chromatic difference meter was calibrated using white and black tiles prior to sample measurement. The L*value was used to determine the degree of roasting, as it is a good indicator of color change during the roasting process and corresponds to the color observations made by the operator.

### 2.3. Preparation of Roasted Fruit Extract

To prepare the roasted fruit extract, 50% ethanol was used as the extraction solvent. The extraction variables were extraction temperature, extraction time, and extraction fluid ratio, which were tested under different conditions ranging from 20 to 60 °C, 15 to 120 min, and 1:10 to 1:30 (*w*:*v*), respectively. Ultrasonic extraction was performed in an ultrasonic bath (JAC-2010; KODO, Seoul, Republic of Korea). The roasted fruit was prepared by roasting at 150 °C for 120 min, and then freeze-drying using a freeze-dryer (FD5508 Bondiro, Ilshin, Republic of Korea). The freeze-dried fruit was ground into a fine powder and used for extraction. The extraction was carried out at 40 °C for 60 min using a solid/liquid ratio of 1:10 (*w*:*v*). The supernatant was obtained by centrifugation (UNION 32R PLUS, Hanil, Korea) at 15,000 rpm for 10 min.

### 2.4. Analysis of DPPH Antioxidant Activity

To evaluate the antioxidant activity, sample solutions (2.5 mL) of different concentrations were mixed with 1 mL of 0.3 mM DPPH ethanol solution and allowed to react at room temperature for 30 min. The absorbance values were measured at 518 nm (SPECTRA MAX 190, Molecular Devices LLC, San Jose, CA, USA), and the percentage of DPPH antioxidant activity was calculated using the following formula: DPPH antioxidant activity (%) = 100 − {[(Abs sample − Abs blank) × 100]/Abs control}. Ethanol (1.0 mL) was used as the blank. Ascorbic acid standard solutions were used as positive controls.

### 2.5. Analysis of Total Polyphenol and Flavonoid Content

Total polyphenol and flavonoid content were measured, as they are known to show a proportional correlation with antioxidant activity. Polyphenol content was determined using the Folin–Ciocalteu method. After roasting, 2 mL of 2% Na_2_CO_3_ solution was added to 100 μL of the sample extracted using 50% ethanol and left for 3 min. Then, 100 μL of 50% Folin–Ciocalteu reagent was added, and the mixture was left at room temperature for 30 min. The absorbance was measured at 750 nm using a UV-Spectrophotometer (SPECTRA MAX 190; Molecular Devices LLC, USA). Gallic acid (Sigma Aldrich, St. Louis, MO, USA) was used as the reference material, and the total polyphenol content was expressed as mg gallic acid in 100 g of the sample. Total flavonoid content was determined by the following method. Then, 250 μL of sample was mixed with 1 mL of distilled water, 75 μL of 5% NaNO_2_, 150 μL of 10% AlCl_3_·6H_2_O, and 500 μL of 1 N NaOH. The mixture was allowed to react at room temperature for 10 min and the absorbance was measured at 510 nm using a UV-Spectrophotometer (SPECTRA MAX 190, Molecular Devices LLC, USA). Quercetin (Sigma-Aldrich, St. Louis, MO, USA) was used as the reference material, and the flavonoid content was expressed as mg quercetin in 100 g of the sample.

### 2.6. MTT Assay

To evaluate the effect of the samples on cell growth, MTT analysis was performed as described by adding MTT solution to form formazan crystals. RAW 264.7 cells were seeded in a 96-well plate at a density of 3 × 10^4^ cells/well in DMEM and cultured for 24 h. After treatment with various concentrations of the samples for 1 h, lipopolysaccharide (LPS; Sigma, USA) was added to a final concentration of 100 ng/mL, and the cells were incubated overnight in a 5% CO_2_ incubator. The culture medium was treated with 3-(4,5-dimethylthazol)-2,5-diphenyltetrazolium bromide (MTT; Amresco, Framingham, MA, USA) for 1 h, followed by the addition of 200 μL of dimethylsulfoxide (DMSO; Amresco, USA). The absorbance was measured at 595 nm using a microplate reader (SPECTRA MAX 190, Molecular Devices LLC, USA) to evaluate the cytotoxicity of the sample.

### 2.7. Levels of ROS

ROS levels were measured using 5-(and-6)-chloromethyl-2′,7′-dichlorodihydrofluorescein diacetate acetate ester (CM-H2DCFDA; Molecular Probes, Eugene, OR, USA). RAW 264.7 cells were seeded in a 96-well plate at a density of 3 × 10^4^ cells/well in DMEM and cultured for 24 h. CM-H2DCFDA was added to a final concentration of 50 μM and cultured for 30 min in a light-shielding state. After washing with PBS, the cells were treated with various concentrations for 1 h. LPS was added to reach a concentration of 500 ng/mL and incubated for 28 h. Fluorescence was measured at a wavelength of 485 nm/520 nm using a fluorescence-measuring device.

### 2.8. Determination of NO Levels 

The murine macrophage cell line RAW 264.7 was used to determine NO levels. The cells were seeded in a 96-well plate at a density of 1 × 10^5^ cells/mL and treated with various sample concentrations for 1 h. After adding 100 ng/mL LPS, the cells were cultured for 24 h, and the generated NO was measured using a Griess reagent. The absorbance was measured at 530 nm, and the amount of NO generated was calculated by comparison with the calibration line of sodium nitrite (NaNO_2_).

### 2.9. β-Hexosaminidase Analysis

RBL-2H3 cells (2 × 10^5^ cells/mL), which are generally described in the literature as a mast cell line were suspended in 10% FBS and DMEM in 24-well plates. The cells were sensitized with dinitrophenyl-ImmunoglobulinE (DNP-IgE) (0.5 μg/mL) and incubated in a 37 °C, 5% CO_2_ incubator for 12 h. The cells were washed twice with Siraganian buffer (119 mM NaCl, 5 mM KCl, 5.6 mM glucose, 0.4 mM MgCl_2_, 25 mM PIPES, 1 mM CaCl_2_, 0.1% BSA, pH 7.2) and then treated with various concentrations of extracts in FBS and DMEM media for 30 min in a 37 °C, 5% CO_2_ incubator. The cells were then stimulated with dinitrophenyl–Human Serum Albumin (DNP-HSA) (2 μg/mL) for 2 h at 37 °C, 5% CO_2_, and the reaction was terminated by cooling in ice. The amount of *β*-hexosaminidase secreted into the supernatant was determined by measuring the absorbance at 405 nm after the reaction with 4-p-nitrophenyl-N-acetyl-D-glucosaminide (2 mM) in sodium citrate buffer (0.05 M, pH 4.5) for 1 h at 37 °C.

### 2.10. Phytochemicals Extraction and Analysis

The 5 g roasted *C. tricuspidata* fruit powders were extracted with 50 mL of methyl alcohol by 100 rpm shaking incubator at 30 °C for overnight. The extract was concentrated to 1/10 using a decompression concentrator and filtered to 0.2 um before HPLC analysis. This filtrate was repeatedly analyzed three times at 280 nm using HPLC (Agilent HPLC (1100 series, Agilent Co., Santa Clara, CA, USA)). Separation was achieved with a LiChrospher RP-18 column (250 mm × 4.6 mm i.d., 5 μm, E. Merck Co., Darmstadt, Germany). The mobile phase consisted of acetonitrile: methanol: water (10:2:88, *v*/*v*/*v*). The flow rate was 1.0 mL/min and the injection volume was 20 μL. Identification of the compounds was carried out by comparing their retention times to those of standards.

### 2.11. Statistical Analysis

Statistical analyses were performed using Student’s *t*-test and Duncan’s multiple range test at a 5% significance level. Correlation analysis of *L, *a, and *b, antioxidant activity (DPPH method), polyphenol content, and flavonoid content was conducted using the R program (version 4.0.2). Data are presented as the mean ± standard deviation (*n* = 3).

## 3. Results and Discussion

### 3.1. Effect of Roasting on the Color of C. tricuspidata Fruits

Consumer acceptance of roasted products is greatly influenced by their color. As depicted in Figure 1, it can be observed that higher roasting temperatures and longer durations led to a reduction in the L*, a*, and b* values. This phenomenon indicates the occurrence of non-enzymatic browning and pyrolysis reactions during the roasting process, resulting in the formation of brown pigments and a darker color in *C. tricuspidata* fruits [15]. Moreover, a decrease in the a* value signifies a decrease in the reddish hue of *C. tricuspidata* fruit, while a decrease in the L* value indicates an increase in darkness. The roasting levels of *C. tricuspidata* fruit ranged from “medium light” to “dark.” These findings align with prior studies conducted on *Pistacia terebinthus* beans [16], hazelnuts [17], and coffee beans [18]. Table 1 provides an overview of the color changes observed in *C. tricuspidata* fruits at different roasting temperatures and durations.

### 3.2. Effect of Roasting Conditions on the DPPH Antioxidant Activity

The color of roasted food items is influenced by various non-enzymatic reactions, including the Maillard reaction and sugar caramelization [19]. In a previous study [20], it was established that the Maillard reaction plays a predominant role during thermal processing in food, leading to the formation of Maillard reaction products. These products are indicative of the degree of thermal treatment and can exhibit diverse biological activities, both beneficial and potentially harmful, such as antioxidant, carcinogenic, and allergenic effects. The objective of this study was to assess the antioxidant activity of roasted *C. tricuspidata* fruit under different conditions by evaluating its DPPH radical scavenging activity.

According to Table 2, the DPPH radical scavenging activity exhibited an increasing trend with higher roasting temperatures and longer durations, ranging from approximately 75% to 93%. These findings align with previous studies conducted on coffee extracts. The enhanced antioxidant activity can be attributed to the formation of Maillard reaction products during the roasting process, as reported in prior research [21]. To support this observation, the polyphenol and flavonoid content of roasted *C. tricuspidata* fruits was measured. Roasting represents a critical heat treatment that significantly impacts food quality in terms of flavor, color, texture, appearance, and composition. It not only eliminates unwanted microorganisms and deactivates enzymes that contribute to product deterioration during storage but also facilitates the formation of antioxidant compounds through non-enzymatic reactions.

### 3.3. Effect of Roasting Conditions on Polyphenol and Flavonoid Content

Polyphenolic compounds, a class of organic molecules commonly present in plants [22], possess notable antioxidant properties that safeguard against oxidative damage by scavenging free radicals and reactive oxygen species [23]. They are also capable of hindering the oxidation of lipids and other molecules by readily donating hydrogen atoms to free radicals [24]. Our study demonstrated an increase in the total polyphenol content in the extracts as the roasting time progressed (Figure 2A). The “untreated” bar refers to fruits that were not subjected to roasting. Significance tests were performed on all bar graphs before and after roasting. The maximum concentration of total polyphenols observed was 169 mg GAE/100 g after 120 min of roasting at 150 °C. While the polyphenol content exhibited an upward trend with increasing temperature and time, it decreased when roasting exceeded 120 min at temperatures above 150 °C. This decrease could be attributed to the natural degradation of phenolic compounds caused by heat treatment. Similar findings have been reported in earlier studies, which observed an increase in polyphenol content in roasted peanuts [25], dry beans [26], and vegetables [27]. It is worth noting that the determination of total polyphenol content involves the analysis of all compounds that react with the Folin–Ciocalteu reagent to form a blue complex, not exclusively polyphenols. Therefore, the observed increase in total polyphenol content in our study could be due to the release of other antioxidant compounds alongside polyphenols. Furthermore, the results may be influenced by the presence of interfering compounds such as formic acid [28]. Additionally, other constituents present in the sample might interfere with the Folin–Ciocalteu reagent and result in an overestimation of the total polyphenol content. However, polyphenols are generally the most abundant antioxidants in plant-based foods, which allows for a rough estimation of the total phenolic compound content [29]. Flavonoids, lignans, and anthocyanins are examples of polyphenolic compounds found in plant-based foods and are known for their antioxidant effects. To assess the antioxidant capacity of roasted *C. tricuspidata* fruits, we quantified their flavonoid content. 

The results presented in Figure 2B show the total flavonoid content of roasted *C. tricuspidata* fruits. Compared to the untreated *C. tricuspidata* fruit, the total flavonoid and polyphenol content of the extract was increased by roasting. Additionally, longer roasting times and higher temperatures positively correlated with an increase in the total flavonoid content of the extract. The untreated *C. tricuspidata* fruit had a total flavonoid content of 2.29 mg QE/100 g, whereas the roasted *C. tricuspidata* fruit had higher total flavonoid content, ranging from 10.98 mg QE/100 g at 110 °C to 16.03 mg QE/100 g at 150 °C for 120 min. The observed increase in flavonoid content can be attributed to the internal tissue disruption caused by heat during the roasting process, which facilitates the extraction of flavonoids [30]. Butera et al. [31] demonstrated that aqueous extracts of prickly pear fruit exhibited significant antioxidant activity due to the presence of betalain pigments. Additionally, Du Toi et al. [32] reported variations in the antioxidant content and activity of cactus pear fruits of different colors, such as purple and pink. The authors noted that purple fruits (*Opuntia robusta* cv *Robusta*) displayed the highest antioxidant content and potential, while orange fruits (*O. ficus-indica*) exhibited the second-highest antioxidant activity, primarily due to elevated levels of ascorbic acid and phenolic compounds. These findings suggest a relationship between fruit color and antioxidant activity. This finding was supported by the significant relationships between DPPH and the color (L*, a*, and b*values) of the roasted *C. tricuspidata* fruits, as shown in Table 3.

Heat treatment has been shown to increase the total flavonoid content in mango and fruits by facilitating the release of phenolic phytochemicals from the internal cells [33]. Similarly, in pineapple juice, heat treatment at 97 °C for 15 min resulted in a higher total flavonoid content compared to fresh juice [34]. Several factors contribute to the generation of new polyphenols and flavonoids during thermal processing, which can be summarized as follows:

Heat-induced cell disruption and deactivation of endogenous oxidative enzymes contribute to an increase in flavonoid content [14]. Furthermore, heat treatment can interfere with plant metabolism, thereby influencing flavonoid levels [35]. Flavonoid content is also influenced by thermal breakdown, where increasing time and temperature can lead to the degradation of flavonoids [36]. Changes in the number and position of hydroxyl groups can occur during heat treatment, resulting in variations in flavonoid content.

In our study, the content of phenolics did not show a correlation with flavonoid content at 15 and 30 min of roasting. Similar findings were reported by Sharma [35], who observed variations in flavonoid content in different types of onions with varying heating temperatures. Additionally, the quercetin glucoside content in red- and brown-skinned onions changed with increasing heating time [37]. In our study, low roasting times may not have released flavonoid-based substances due to the fibrous separation of the fruits.

### 3.4. Relationships Amongst Color, DPPH, Polyphenol Content, and Flavonoid Content of Roasted C. tricuspidata Fruit

Figure 3 displays the correlation analysis results for *L, *a, *b, DPPH antioxidant activity, polyphenol content, and flavonoid content of roasted *C. tricuspidata* fruits. Our findings demonstrate a strong correlation between DPPH and polyphenol content, which is consistent with previous studies [38]. Phenolic compounds, which are well-known plant antioxidants, can donate hydrogen atoms to free radicals, thereby contributing to their antioxidant capacities [39]. Our study also revealed a high correlation between DPPH and polyphenol content (r = 0.90, *p* < 0.001), which is consistent with previous findings in various fruits [40]. Our results further revealed a strong negative correlation between *L (r = –0.68), *a (r = −0.79), and *b (r = −0.80) parameters and DPPH antioxidant activity, indicating that as the color of the roasted fruit darkens, the antioxidant activity increases. This trend is supported by recent studies that have reported that legumes with darker colors have high antioxidant activity due to the presence of phenolic compounds, mainly anthocyanins and condensed tannins [41]. Finally, we assessed the anti-inflammatory efficacy of *C. tricuspidata* fruits roasted for 120 min at 150 °C, which exhibited the highest antioxidant activity.

### 3.5. Inhibitory Effect of Roasted C. tricuspidata Fruits on ROS and NO Production

In the present study, the antioxidant activity of roasted *C. tricuspidata* fruits was investigated by evaluating their anti-inflammatory effects. The cell viability of the roasted fruit was tested and no significant cytotoxicity was observed up to 1000 mg/L (Figure 4A). To determine whether the roasted fruit inhibited ROS and NO production, LPS-induced RAW 264.7 macrophages were treated with various concentrations of the roasted fruit, as well as non-roasted *C. tricuspidata* fruit to compare the effect of roasting. Both roasted and non-roasted fruits significantly inhibited LPS-induced NO and ROS production in a dose-dependent manner (Figure 4B,C, *p* < 0.001). Roasted fruit was particularly effective at inhibiting NO and ROS production compared to non-roasted fruit. This study also discussed the potential role of phenolic compounds in preventing human diseases through their antiradical activity, which involves the donation of hydrogen atoms from aromatic hydroxyl groups to free radicals [42]. Free radicals, including superoxide (O_2_^−^), hydroxyl (OH^−^), peroxyl (ROO), and nitric oxide (NO) radicals, are involved in biological metabolism [43]. However, an imbalance between free radicals and antioxidants leads to oxidative stress, which is a major cause of various chronic diseases [44]. The polyphenols in roasted *C. tricuspidata* fruit were well-documented in this study because of their significant inhibitory effect on ROS and NO production, suggesting that these compounds may play a significant role in preventing chronic diseases.

### 3.6. Effect of Roasted C. tricuspidata Fruit on β-Hexosaminidase in RBL-2H3 Cells

*β*-hexosaminidase is a cytoplasmic granule enzyme found in mast cells, and its activation leads to degranulation. In the present study, roasted *C. tricuspidata* fruit significantly suppressed the degranulation of DNP-BSA-stimulated RBL-2H3 cells (Figure 5B). Furthermore, the inhibitory effect of roasted fruit on cell degranulation was dose dependent. Compared to quercetin, which was used as a positive control, roasted *C. tricuspidata* fruit showed higher levels of *β*-hexosaminidase but significantly reduced *β*-hexosaminidase release in DNP-BSA-stimulated RBL-2H3 cells. All tested concentrations of roasted *C. tricuspidata* fruit inhibited *β*-hexosaminidase release, with a significant difference of *p* < 0.001. These findings suggest that roasted *C. tricuspidata* fruit may have stronger anti-allergic effects on RBL-2H3 cells than non-roasted *C. tricuspidata* fruit (Figure 5A).

### 3.7. Changes in Phytochemical Content of Fruits Caused by Roasting

Phytochemical compounds found in plants are well-known for their antioxidant properties and ability to scavenge free radicals, which can have potential health benefits for humans [45]. In our study, we aimed to investigate the composition of phytochemical compounds in both unroasted and roasted fruits by conducting HPLC analysis on 80% methanol extracts. Table 3 presents the identified phytochemicals and their corresponding content. Our hypothesis was that the increase in phenolic content observed in roasted fruits could be attributed to the release of phenolic acids from the cell matrix during thermal processing. The breakdown of cell membranes and walls during roasting allows for the release of soluble phenolic compounds, leading to an increase in solubilized phenolic content and consequently enhancing the antioxidant activity of *C. tricuspidata* fruit during the roasting process. Previous studies on vegetables have demonstrated that increased solubilized ferulic acid content contributes to an increase in total antioxidant activity, particularly at high temperatures [46]. Similarly, in a previous study [47], it was found that solubilization of chlorogenic acid from small black soybeans occurs only at temperatures exceeding 150 °C, resulting in an increase in total antioxidant activity. Our findings align with these previous observations, as we observed a significant increase in antioxidant activity at temperatures of 150 °C.

Furthermore, edible plants are known to contain flavan-3-ols, flavanols, and phenolic acids, which offer various health benefits [48]. Flavan-3-ols such as catechin and epicatechin have been reported to possess antiviral, anticarcinogenic, and hypocholesterolemic activities [49]. Phenolic acids, including gallic acid and its derivatives, exhibit antioxidant and anticarcinogenic properties, and their daily intake has been associated with various health benefits, including a reduced risk of diseases [50]. Additionally, a previous study [51] reported that *C. tricuspidata* leaves are rich in phytochemicals such as quercetin, gallic acid, catechin, and chlorogenic acid, which display strong antioxidant activity and radical scavenging effects. These findings suggest that the increase in flavan-3-ol and phenolic acid content may contribute to the enhanced antioxidant activity observed in roasted *C. tricuspidata* fruits.

## 4. Conclusions

In conclusion, this study aimed to investigate the polyphenol content and antioxidant and anti-inflammatory activities of *C. tricuspidata* fruit extracts obtained using roasting and non-roasting methods. Results showed that the roasted *C. tricuspidata* fruit exhibited higher DPPH radical-scavenging activity in a temperature- and time-dependent manner. Additionally, the selected roasted *C. tricuspidata* fruit extracts showed potent anti-inflammatory properties in LPS-induced RAW 264.7 macrophages at non-toxic concentrations, as evidenced by their ability to inhibit ROS and NO production. Interestingly, the color change of roasted *C. tricuspidata* fruit was strongly correlated with both DPPH scavenging activity and anti-inflammatory effects. These findings suggest that roasted *C. tricuspidata* fruits, rich in polyphenols, have potential health-promoting benefits. Interestingly, the increase in antioxidant activity was attributed to the increase in flavan-3-ols and phenolic acids in the roasted *C. tricuspidata* fruits. Overall, this study contributes to the growing body of literature on the health benefits of *C. tricuspidata* fruit and its potential application as a functional food ingredient. To our knowledge, no phytochemical content of roasted *C. tricuspidata* fruits has been reported, and our study is the first.

## Figures and Tables

**Figure 1 foods-12-02146-f001:**
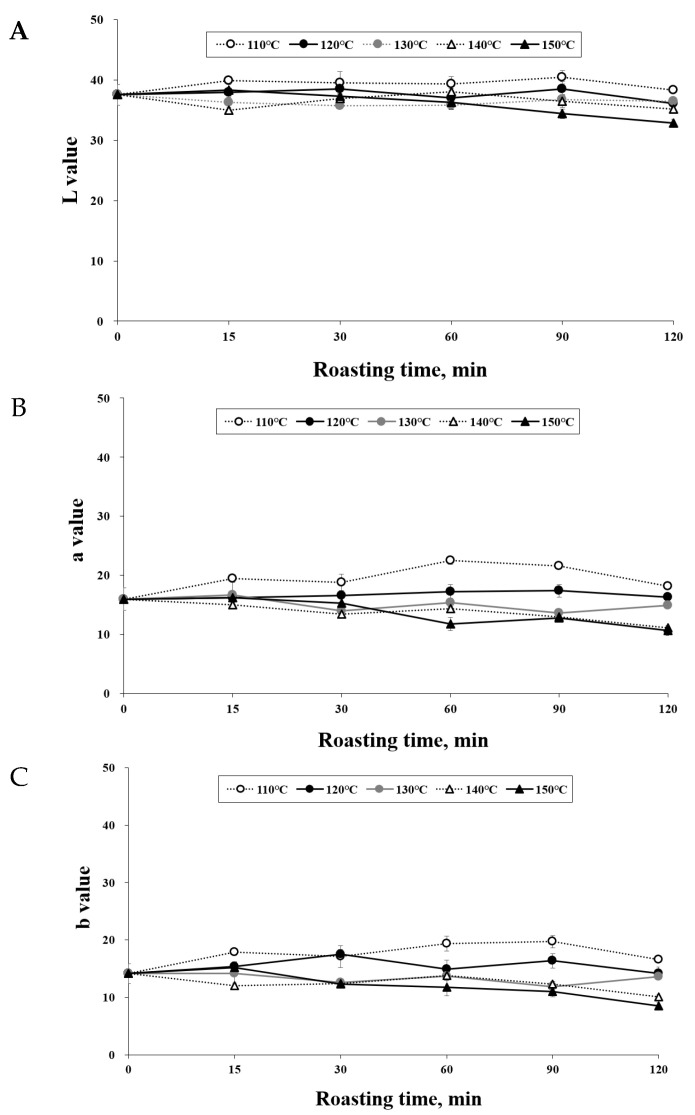
Changes in chromaticity values of roasted *C. tricuspidata* fruit. (**A**): L* value of roasted *C. tricuspidata* fruit; (**B**): a* value of *C. tricuspidata* fruit; (**C**): b* value of *C. tricuspidata* fruit.

**Figure 2 foods-12-02146-f002:**
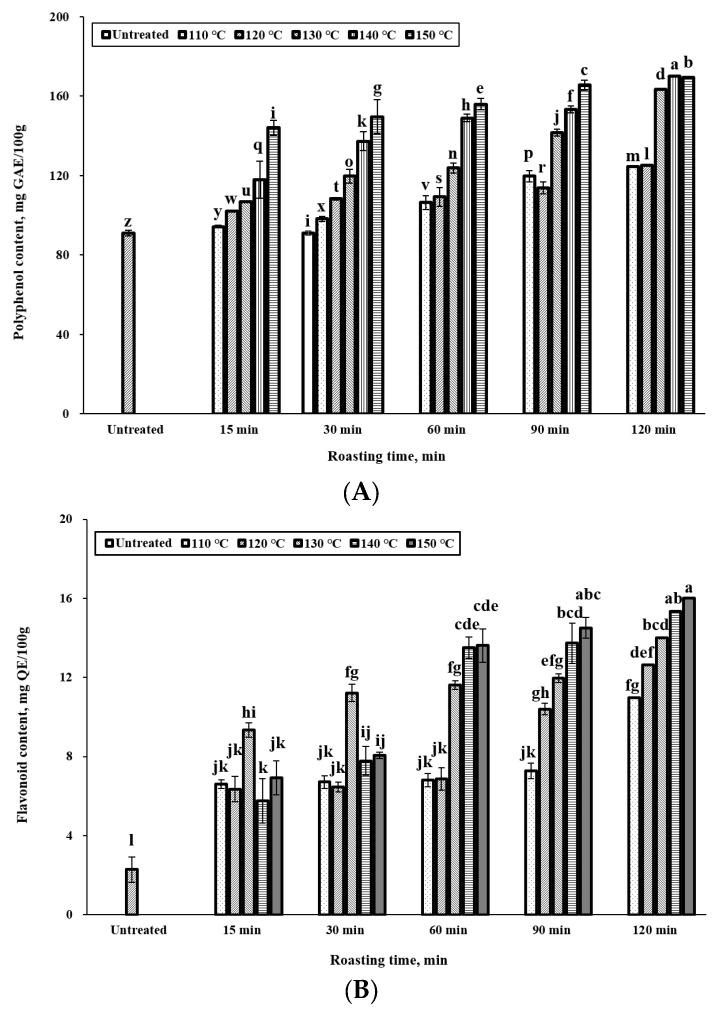
Effect of roasting condition on polyphenol content and flavonoid content of *C. tricuspidata* fruit. (**A**): polyphenol content; (**B**): flavonoid content. Different lowercase letters represent significant statistical difference (*p* < 0.05) between the different roasting conditions or the untreated group.

**Figure 3 foods-12-02146-f003:**
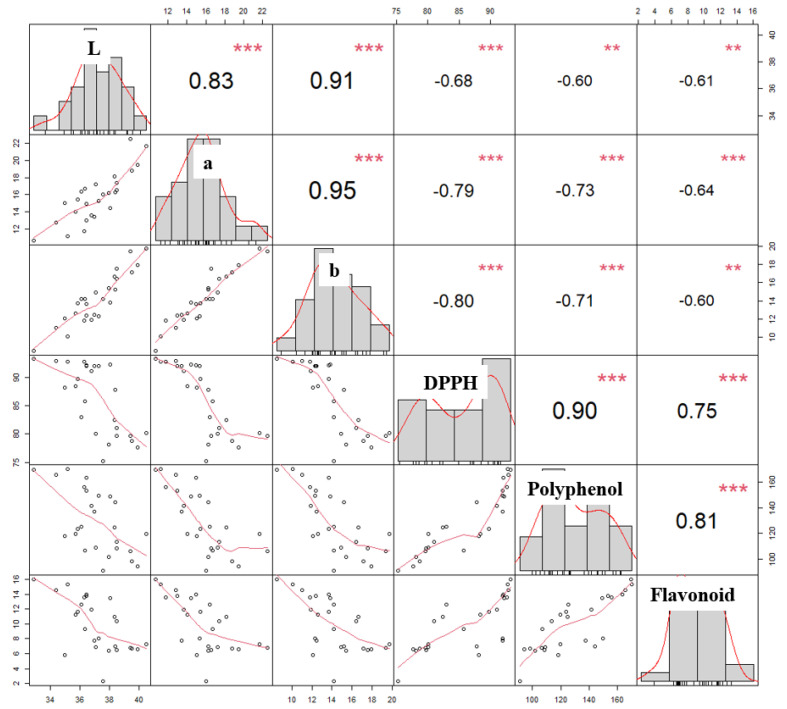
Correlations between fruit color (*L, *a, and *b), DPPH, polyphenol content, and flavonoid content. The distribution of each variable is shown on the diagonal. On the bottom of the diagonal: the bivariate scatter plots with a fitted line are displayed. On the top of the diagonal: the value of the correlation plus the significance level as stars. Each significance level is associated to a symbol: *p*-values (0.001, 0.01, 0.05) <=> symbols (“***”, “**”).

**Figure 4 foods-12-02146-f004:**
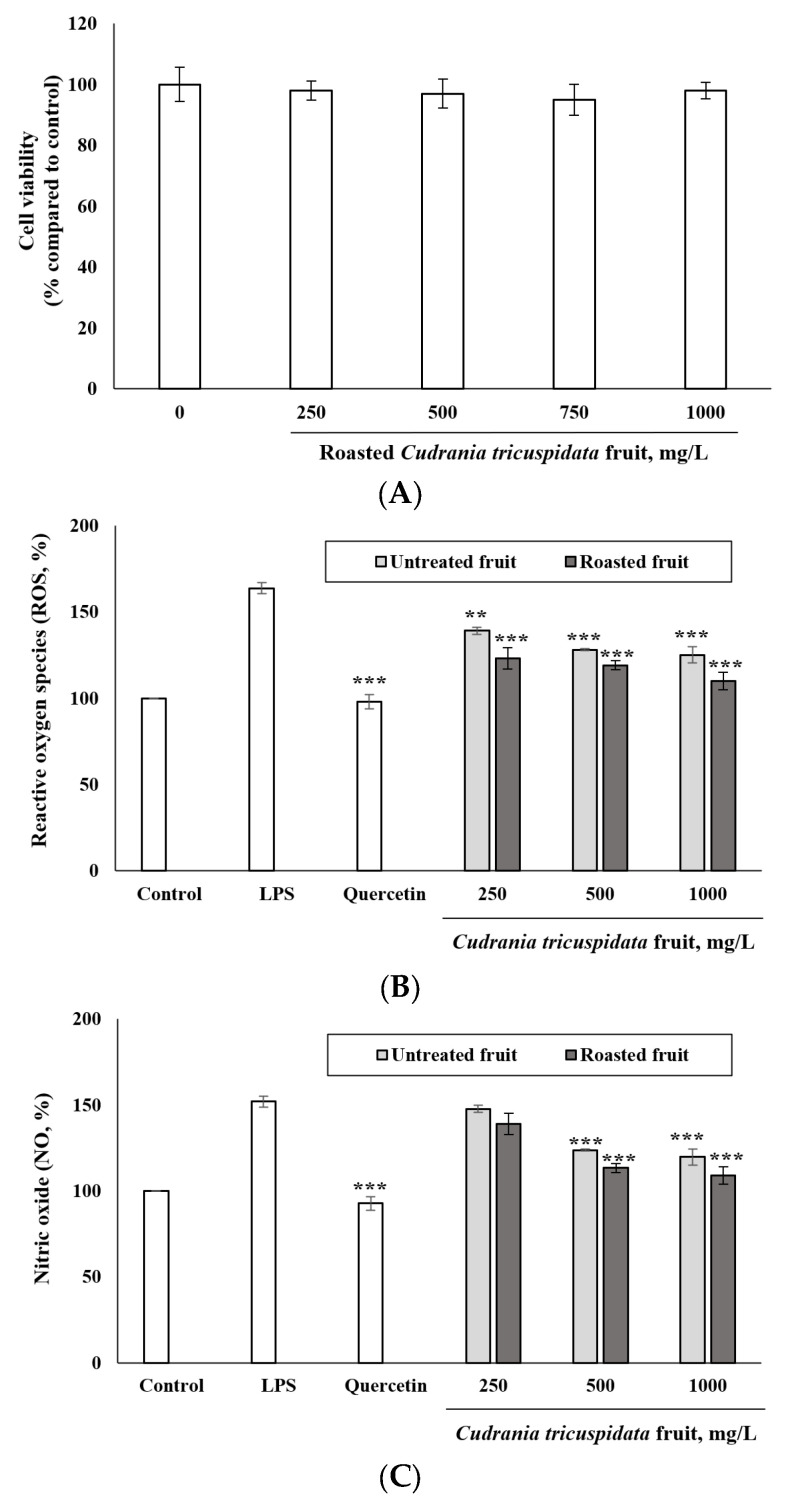
Effects of *C. tricuspidata* fruit on cell viability (**A**), ROS (**B**) and NO (**C**) production in LPS-induced macrophage RAW 264.7 cells. ** *p* < 0.01 and *** *p* < 0.001 compared to each LPS alone.

**Figure 5 foods-12-02146-f005:**
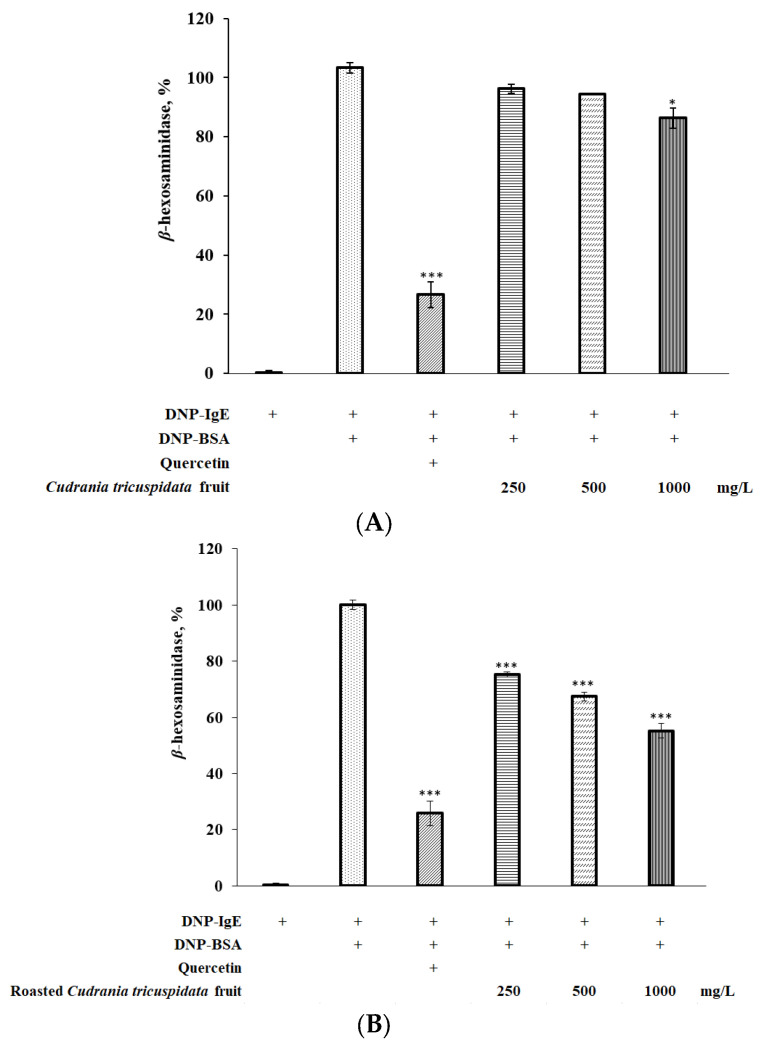
Effect of roasting on *β*-hexosaminidase inhibition of *C. tricuspidata* fruit. (**A**): non-roasted *C. tricuspidata* fruit; (**B**): roasted *C. tricuspidata* fruit. * *p* < 0.05; *** *p* < 0.001 compared DNP-IgE + DNP-BSA.

**Table 1 foods-12-02146-t001:** Color changes of roasted *C. tricuspidata* fruits at different roasting temperatures and times.

Temp., °C	Time, min				
	15	30	60	90	120
110	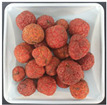	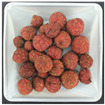	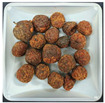	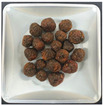	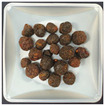
120	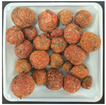	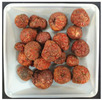	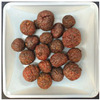	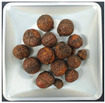	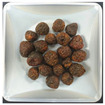
130	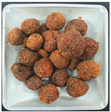	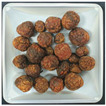	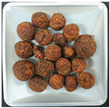	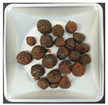	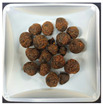
140	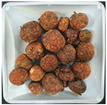	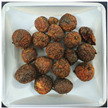	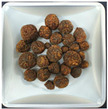	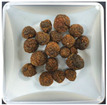	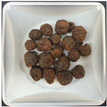
150	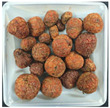	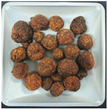	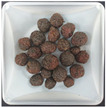	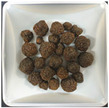	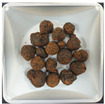

**Table 2 foods-12-02146-t002:** Experimental data of DPPH antioxidant activity of *C. tricuspidata* fruit as affected by the roasting conditions.

T (°C)	T (min)	DPPH Antioxidant Activity
Untreated	-	75.29 ± 0.28 ^e (1)^
110	15	77.66 ± 0.64 ^d^
	30	78.73 ± 1.18 ^ed^
	60	79.68 ± 0.55 ^c^
	90	80.14 ± 0.41 ^c^
	120	82.42 ± 1.54 ^b^
120	15	78.17 ± 1.56 ^d^
	30	79.74 ± 1.17 ^cd^
	60	80.05 ± 0.87 ^cd^
	90	81.02 ± 0.39 ^c^
	120	82.97 ± 1.67 ^b^
130	15	85.72 ± 5.92 ^c^
	30	88.40 ± 4.74 ^bc^
	60	89.72 ± 3.45 ^bc^
	90	91.04 ± 1.56 ^bc^
	120	92.11 ± 1.12 ^b^
140	15	88.17 ± 6.00 ^c^
	30	91.97 ± 0.45 ^bc^
	60	92.21 ± 0.62 ^bc^
	90	91.92 ± 1.43 ^bc^
	120	92.73 ± 0.39 ^b^
150	15	87.82 ± 6.35 ^c^
	30	91.94 ± 0.55 ^bc^
	60	92.67 ± 0.73 ^b^
	90	92.83 ± 0.70 ^b^
	120	93.22 ± 0.41 ^b^
Ascorbic acid	-	98.18 ± 0.06 ^a^

^(1)^ Mean values ± SD of determination for triplicate samples; means followed by similar columns superscript in the same column are not different (*p* < 0.05).

**Table 3 foods-12-02146-t003:** Distributions on phytochemical content of *C. tricuspidata* fruit by roast processing.

Phytochemical Content, μg/g	Unroasted	Roasted ^(1)^	Class
Epigallocatechin	400.1 ± 3.2 ^(2)^	568.2 ± 2.5	Flavan-3-ol derivatives
Catechin	311.8 ± 3.8	452.6 ± 3.5	Flavan-3-ol derivatives
Epicatechin	110.7 ± 2.5	284.3 ± 3.0	Flavan-3-ol derivatives
Gallic acid	66.5 ± 1.5	97.1 ± 3.4	Phenolic acid derivatives
Tannic acid	27.8 ± 0.7	46.6 ± 2.6	Phenolic acid derivatives
Vanillic acid	68.4 ± 1.2	95.2 ± 1.5	Phenolic acid derivatives
Caffeic acid	64.1 ± 0.1	92.2 ± 1.7	Phenolic acid derivatives
*p*-Coumaric acid	44.9 ± 1.7	58.2 ± 0.1	Phenolic acid derivatives
Ferulic acid	38.9 ± 2.6	43.5 ± 0.2	Phenolic acid derivatives
Rutin	47.5 ± 0.2	64.7 ± 1.2	Flavanol derivatives
Quercetin	20.1 ± 0.0	31.9 ± 0.1	Flavanol derivatives
Kaempferol	13.9 ± 0.1	2.4 ± 0.0	Flavanol derivatives

^(1)^ *C. tricuspidata* fruit was roasted at 150 °C 120 min. ^(2)^ All values are presented as the mean ± SD of triplicate determinations.

## Data Availability

Data are contained within the article.

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
