# Peer review of "Changes in Phytochemical Content, Antioxidant Activity, and Anti-Inflammatory Properties of Cudrania tricuspidata Fruits Treated by Roasting"

_foods, 2023, doi:10.3390/foods12112146_

Round 1
Reviewer 1 Report
The authors prepared interesting paper concerning the differences in biological activities of fresh and roasted fruits of C. tricuspidata. Although it brings some new information, several issues should be address befor publishing:
Major issues:
1.) Chapter 3.3. - What is the source of "new" polyphenols and flavonoids? Prepared and tested samples were lyophilized and grinded, hence, they contained both extracellular and intracellular matrix. Is it caused by heat destruction of some polymer? Or oxidation of phytochemicals? It should be discuss in better way.
2.) Fig. 2 - The content of phenolics does not correlate with flavonoid content at 15 and 30 min roasting. It should be discuss.
Minor issue:
1.) Tab. 2. - What do mean letters behind numbers? It should be describe in the caption.
Author Response
Manuscript ID: foods-2390432
Response to Reviewers
Dear Journal of “Foods”,
Thank you for giving us the opportunity to submit a revised version of the manuscript entitled “Changes in phytochemical content, antioxidant activity and anti-inflammatory of Cudrania tricuspidata fruits treated by roasting” for publication in the Journal of “Foods”. We appreciate the time and effort invested by you and the reviewers to provide insightful feedback regarding our manuscript, which has helped us make valuable improvements to our paper. We have incorporated most of the suggestions. These changes have been highlighted in the manuscript. We have responded to each of the reviewers’ comments below. All page numbers pertain to the revised manuscript file with tracked changes visible inline.
Reviewers' Comments to the Authors:
Reviewer 1
[ Major issues ]
- Chapter 3.3. - What is the source of "new" polyphenols and flavonoids? Prepared and tested samples were lyophilized and grinded, hence, they contained both extracellular and intracellular matrix. Is it caused by heat destruction of some polymer? Or oxidation of phytochemicals? It should be discuss in better way.
Author response: Thank you for pointing this out.
The revised text reads as follows on page 8, line 247:
“Heat treatment was found to result in higher total flavonoid content in mango and ber fruits, as it facilitates the release of phenolic phytochemicals from internal cells [33]. Additionally, heat treatment (at 97℃ for 15 minutes) led to a higher total flavonoid con-tent in pineapple juice compared to fresh juice [34]. Several factors contribute to the gener-ation of new polyphenols and flavonoids, which can be summarized as follows:
(i) Heating disrupts cells and deactivates endogenous oxidative enzymes, leading to an increase in flavonoid content [35]. Moreover, heat treatment may also interfere with plant metabolism, thereby influencing flavonoid content [36].
(ii) Flavonoid content is further affected during thermal processing due to the break-down of flavonoids with increasing time and temperature [37].
(iii) The number and position of hydroxyl groups can undergo changes during ther-mal treatment, resulting in variations in flavonoid content.”
The added reference was showed on page 15, line 446:
- Kavitha, C.; Kuna, A. Effect of processing on antioxidant properties of ber (Zizyphus mauritiana) Fruit. Int. J. Sci. Res. 2014, 3, 2019–2025. https://www.ijsr.net/archive/v3i7/MDIwMTU3MA==.pdf
- Goh, S.G.; Noranizan, M.; Leong, C.M.; Sew, C.C.; Sobhi, B. Effect of thermal and ultraviolet treatments on the stability of antioxidant compounds in single strength pineapple juice throughout refrigerated storage. Int. Food Res. J. 2012, 19, 1131–1136. http://ifrj.upm.edu.my/19%20(03)%202012/(46)%20IFRJ%2019%20(03)%202012%20Nora.pdf
- Choi, Y.; Lee, S.M.; Chun, J.; Lee, H.B.; Lee, J. Influence of heat treatment on the antioxidant activities and polyphenolic compounds of Shiitake (Lentinus edodes) mushroom. Food Chem. 2006, 99, 381–387. https://doi.org/10.1016/j.foodchem.2005.08.004
- Sharma, K. Temperature-dependent studies on the total phenolics, flavonoids, antioxidant activities, and sugar content in six onion varieties. J. Food Drug Anal. 2015, 23, 243–252. https://doi.org/10.1016/j.jfda.2014.10.005
- Mohd Zainol, M.K.; Abdul-Hamid, A.; Abu Bakar, F.; Pak Dek, S. Effect of different drying methods on the degradation of selected flavonoids in Centella asiatica. Int. Food Res. J. 2009, 16, 531–537.
- Fig. 2 - The content of phenolics does not correlate with flavonoid content at 15 and 30 min roasting. It should be discuss.
The revised text reads as follows on page 8, line 259:
“In our results, the content of phenolics does not correlate with flavonoid content at 15 and 30 min roasting. Sharma [38] reported that flavonoid content in different onions can vary with heating temperature. Quercetin glucoside content in red- and brown-skinned onions changed with the increase in heating time [39]. Our study also would not have released flavonoid-based substances due to fibrous separation of fruits at low roasting times.”
The added reference was showed on page 15, line 457:
- Sharma, K. Temperature-dependent studies on the total phenolics, flavonoids, antioxidant activities, and sugar content in six onion varieties. J. Food Drug Anal. 2015, 23, 243–252. https://doi.org/10.1016/j.jfda.2014.10.005
- Price, K.R.; Bacon, J.R.; Rhodes, M.J.C. Effect of storage and domestic processing on the content and composition of flavonol glucosides in onion (Allium cepa). J. Agric. Food Chem. 1997, 45, 938–942. https://doi.org/10.1021/jf9605916
[ Minor issue ]
- Tab. 2. - What do mean letters behind numbers? It should be describe in the caption.
The revised text reads as follows on Table 2:
“1)Mean values ± SD of determination for triplicate samples; means followed by similar columns superscript in the same column are not different (p < 0.05).”

Reviewer 2 Report
1. The scientific name should be italic throughout the manuscript. Check the scientific name from this website : http://plantlist.org/
2. The abstract should be improved.
3. Add the accession number of the plant.
4. Discussion section should be added.
5.In vivo anti-inflammatory activity should be assessed. Protocol can be found from these article: https://doi.org/10.1016/j.jsps.2020.11.004
6. Evaluate the data through molecular modelling too to justify and validate the lab findings.
7. There are few typos as well.
Can be improved. Few grammatical mistakes as well as typos are in the manuscript.
Author Response
Manuscript ID: foods-2390432
Response to Reviewers
Dear Journal of “Foods”,
Thank you for giving us the opportunity to submit a revised version of the manuscript entitled “Changes in phytochemical content, antioxidant activity and anti-inflammatory of Cudrania tricuspidata fruits treated by roasting” for publication in the Journal of “Foods”. We appreciate the time and effort invested by you and the reviewers to provide insightful feedback regarding our manuscript, which has helped us make valuable improvements to our paper. We have incorporated most of the suggestions. These changes have been highlighted in the manuscript. We have responded to each of the reviewers’ comments below. All page numbers pertain to the revised manuscript file with tracked changes visible inline.
Reviewers' Comments to the Authors:
Reviewer 2
[ Major issues ]
- The scientific name should be italic throughout the manuscript. Check the scientific name from this website : http://plantlist.org/
Author response: Thank you for pointing this out.
The revised text reads as follows on Title:
“Changes in phytochemical content, antioxidant activity and anti-inflammatory of Cudrania tricuspidata fruits treated by roasting”
The revised reference was showed on page 14, line 388:
- Jin, Q.; Yang, J.; Ma, L.; Cai, J.; Li, J. Comparison of polyphenol profile and inhibitory activities against oxidation and α‐glucosidase in mulberry (Genus Morus) cultivars from China. J. Food Sci. 2015, 80, C2440-C2451. https://doi.org/10.1111/1750-3841.13099
- The abstract should be improved.
Author response: Thank you for pointing this out.
The revised text reads as follows on page 1, line 9:
“The study investigated the antioxidant effects of roasted Cudrania tricuspidata (C. tricuspidata) fruits by comparing them with unroasted C. tricuspidata fruits. The results showed that the roasted C. tricuspidata fruits (150℃, 120 min) exhibited significantly higher antioxidant activity, especially in terms of anti-inflammatory effects, than the unroasted fruits. Interestingly, there is a high correlation between the color of the roasted fruit and the antioxidant activity. Heating disrupts cells and deactivates endogenous oxidative enzymes, leading to an increase in flavonoid content. Moreover, heat treatment may also interfere with plant metabolism, thereby influencing flavonoid content. Moreover, HPLC analysis of roasted fruits in our study shows that, the increase in antioxidant activity was at-tributed to the increase in flavan-3-ols and phenolic acids in the roasted C. tricuspidata fruits. To the best of our knowledge, it is the first time to study the antioxidant activity and anti-inflammation of roasted C. tricuspidata fruits. The study concluded that roasted C. tricuspidata fruits could be a valuable natural source of antioxidants for various food and medicinal applications.”
- Add the accession number of the plant.
Author response: Thank you for pointing this out.
We have not deposited it with Genbank yet, and we will proceed with the deposit as soon as possible.
- Discussion section should be added.
Author response: Thank you for pointing this out.
We wrote the "Results and Discussion" section according to the general form published in the "Foods" journal.
- In vivo anti-inflammatory activity should be assessed. Protocol can be found from these article: https://doi.org/10.1016/j.jsps.2020.11.004
Author response: Thank you for pointing this out.
We used the 'in vitro' experiment to study anti-inflammatory, not the 'in vivo' experiment. We thought that the word 'intracellular' that we used was misleading. So I deleted all the words 'intracular' from the text.
- Evaluate the data through molecular modelling too to justify and validate the lab findings.
Author response: Thank you for pointing this out.
Molecular modelling encompasses all methods, theoretical and computational, used to model or mimic the behaviour of molecules. The methods are used in the fields of computational chemistry, drug design, computational biology and materials science to study molecular systems ranging from small chemical systems to large biological molecules and material assemblies. The simplest calculations can be performed by hand, but inevitably computers are required to perform molecular modelling of any reasonably sized system. The common feature of molecular modelling methods is the atomistic level description of the molecular systems. Although this study focused on antioxidant, anti-inflammatory, and component analysis of roasted fruits, we will conduct research on molecular modeling in the future and post it as a new paper. Thank you for your careful consideration.
- There are few typos as well.
Author response: Thank you for pointing this out.

Reviewer 3 Report
Dear authors,
Thank you for a well written manuscript. A few comments that I feel will improve the manuscript:
1. For a well written manuscript the abstract is a bit of a let down. Its too brief and, in some cases, confusing e.g., ‘C. tricuspidata fruits exhibited significantly higher antioxidant activity, especially in terms of anti-inflammatory effects’ Do the authors mean higher antioxidant activity correlated well with higher anti-inflammatory activity?
2. Section 2.9 – what cells are these (RBL-2H3)?
3. Section 2.11 – were normality tests done before checking for significance?
4. Figure 2A – 30 min has 6 bars instead of 5. What’s the extra bar?
5. Figure 2A and 2B – are the comparisons done in relation to the untreated? If so, why does the untreated have a significance letter? Unless these values were ranked? Please clarify.
6. Figure 3 is confusing – is there a way to relabel where one can see e.g., which correlation is between polyphenol content vs. DPPH.
Author Response
Manuscript ID: foods-2390432
Response to Reviewers
Dear Journal of “Foods”,
Thank you for giving us the opportunity to submit a revised version of the manuscript entitled “Changes in phytochemical content, antioxidant activity and anti-inflammatory of Cudrania tricuspidata fruits treated by roasting” for publication in the Journal of “Foods”. We appreciate the time and effort invested by you and the reviewers to provide insightful feedback regarding our manuscript, which has helped us make valuable improvements to our paper. We have incorporated most of the suggestions. These changes have been highlighted in the manuscript. We have responded to each of the reviewers’ comments below. All page numbers pertain to the revised manuscript file with tracked changes visible inline.
Reviewers' Comments to the Authors:
Reviewer 3
[ Major issues ]
- For a well written manuscript the abstract is a bit of a let down. Its too brief and, in some cases, confusing e.g., ‘C. tricuspidata fruits exhibited significantly higher antioxidant activity, especially in terms of anti-inflammatory effects’ Do the authors mean higher antioxidant activity correlated well with higher anti-inflammatory activity?
Author response: Thank you for pointing this out.
The revised text reads as follows on page 1, line 9:
“The study investigated the antioxidant effects of roasted Cudrania tricuspidata (C. tricuspidata) fruits by comparing them with unroasted C. tricuspidata fruits. The results showed that the roasted C. tricuspidata fruits (150℃, 120 min) exhibited significantly higher antioxidant activity, especially in terms of anti-inflammatory effects, than the unroasted fruits. Interestingly, there is a high correlation between the color of the roasted fruit and the antioxidant activity. Heating disrupts cells and deactivates endogenous oxidative enzymes, leading to an increase in flavonoid content. Moreover, heat treatment may also interfere with plant metabolism, thereby influencing flavonoid content. Moreover, HPLC analysis of roasted fruits in our study shows that, the increase in antioxidant activity was at-tributed to the increase in flavan-3-ols and phenolic acids in the roasted C. tricuspidata fruits. To the best of our knowledge, it is the first time to study the antioxidant activity and anti-inflammation of roasted C. tricuspidata fruits. The study concluded that roasted C. tricuspidata fruits could be a valuable natural source of antioxidants for various food and medicinal applications.”
- Section 2.9 – what cells are these (RBL-2H3)?
Author response: Thank you for pointing this out.
The revised text reads as follows on page 4, line 142:
“RBL-2H3 cells (2 × 105 cells/mL), which is generally described in the literature as a mast cell line were suspended in 10% FBS and DMEM in 24-well plates.”
- Section 2.11 – were normality tests done before checking for significance?
Author response: Thank you for pointing this out.
All significance tests were obtained after normality analysis.
- Figure 2A – 30 min has 6 bars instead of 5. What’s the extra bar?
Author response: Thank you for pointing this out.
The revised text reads as follows on page 7, line 210:
“The "untreated" bar means fruit not roasted.” I have made additional comments on this in the text as well.
- Figure 2A and 2B – are the comparisons done in relation to the untreated? If so, why does the untreated have a significance letter? Unless these values were ranked? Please clarify.
Author response: Thank you for pointing this out.
The revised text reads as follows on page 7, line 210:
“All bar graphs were analyzed as significance tests before and after roasting.”
- Figure 3 is confusing – is there a way to relabel where one can see e.g., which correlation is between polyphenol content vs. DPPH.
Author response: Thank you for pointing this out.
I drew more clearly about the factor.

Round 2
Reviewer 1 Report
The authors addressed all mentioned issues.
Author Response
You said, "The authors addressed all mentioned issues." Thank you again for your review. I am sure that the quality of my paper has improved thanks to your review. Thank you.
Reviewer 2 Report
Accession number is required for the authenticity of the plant sample. The accession number can be availed from national herbarium or other regulatory authorities by depositing the plant sample. Without the accession number of plant sample, the claim can sound unreliable. So please add this in the manuscript.
For example, check the 2.1 no paragraph. You can see the accession number inclusion style.
Article: Pharmacological and computer aiddd studies provide new insights into Millettia peguensis Ali.
Its fine
Author Response
Thank you for pointing this out.
1. Accession number is required for the authenticity of the plant sample.
The revised reference was showed on 2 page 69 line
I was able to solve this problem through your kind example. The revised part is as follows.
"The C. tricuspidata were ascertained by the experts of Korea National Arboretum and a voucher specimen (Accession no: CJUA200909151089) has been deposited for this collection. "
I'm sure your review helped me improve the quality of my paper. Thanks once again.